# Long-Term (>10 Years) Effects of Medical and Surgical Airway Obstruction Treatment on Dentofacial Morphology

**DOI:** 10.3390/diagnostics15233079

**Published:** 2025-12-03

**Authors:** Anthony T. Macari, Annie Babakhanian, Ingrid Karam, Joseph G. Ghafari

**Affiliations:** 1Division of Orthodontics and Dentofacial Orthopedics, Department of Dentofacial Medicine, Faculty of Medicine, American University of Beirut, Beirut 1107 2020, Lebanon; ingridkaram@gmail.com (I.K.); jg03@aub.edu.lb (J.G.G.); 2Department of Orthodontics, School of Dental Medicine, University of Pennsylvania, Philadelphia, PA 19104, USA

**Keywords:** mouth breathing, adenoidectomy, cephalometry, craniofacial abnormalities, airway obstruction

## Abstract

Airway obstruction may lead to dentofacial dysmorphogenesis, with severity influenced by age, duration, and extent of obstruction. **Aims:** to evaluate long-term craniofacial changes in children with a history of mouth breathing, comparing outcomes between those treated with lymphoid tissue removal or with medication, and considering treatment age. **Materials and Methods:** Fifty-seven patients with a mean age of 19.09 years (range: 15.1–25.2 years) who had been evaluated in an earlier study (T1) were recalled at an average of 13 years follow-up (T2) and classified into a surgical group (n = 34), who had an adenoidectomy, and a non-surgical group (n = 23) treated with medication. Lateral cephalograms were obtained and compared with the original pre-treatment records. Control groups were included, matching the subjects in both groups for age and sex. Statistical analyses included group comparisons and associations among variables. **Results:** Significant improvement in both treatment groups were observed for the gonial angle (Ar-Go-Me), facial convexity (S-N-Me) and facial height (N-Gn), but T2-T1 changes in the surgical group were statistically significantly greater than in the medical therapy group. The palatal plane inclination to the horizontal (PP/H) and the mandibular plane inclination (MP/H) and to cranial base (MP/SN) were significantly improved in both groups (0.001 < *p* < 0.01). **Conclusions:** Both surgical and medical treatment of airway obstruction resulted in the reversal of the harmful effect of the obstruction. However, adenoidectomy was associated with greater improvements, possibly because the original obstruction was more severe and longer-standing. The results underline the importance of early recognition and management of airway obstruction to mitigate developmental orofacial dysmorphology.

## 1. Introduction

The association between malocclusion and impaired airways requires the assessment of abnormal respiration and its potential impact on the shape and size of jaws and dentition [1,2,3,4,5,6]. Orofacial adaptation to mouth breathing and the extent of the ensuing morphological changes depend on the onset and duration of the insult and gravity of airway obstruction, most severely revealed in the long-face syndrome or “adenoid facies” [7]. Accordingly, these characteristics impact timing, management, and stability of orthodontic treatment.

Airway obstruction is reportedly more acute at a younger age, warranting early examinations to detect and intercept developing dentofacial malformations [8]. The harmful effects of the obstruction on facial features may be reversed with airway clearance. The post-surgical effects of adenoidectomy/tonsillectomy on dentofacial morphology include more anterior symphyseal growth, reversal of the tendency to posterior mandibular rotation, and increased mandibular growth [9,10,11,12,13,14]. Follow-up studies ranged between one to five years post-adenoidectomy/tonsillectomy, [11,12,13,14,15,16,17] but knowledge of longer-term sustenance of those changes is needed.

Medication (mainly intranasal corticosteroids) is commonly the first intervention in the treatment of adenoid hypertrophy, particularly when mild or moderate, and often provisionally before committing to surgery. Adenoidectomy provides immediate relief by physically removing the obstructive lymphoid tissue, thus restoring nasal breathing promptly. In contrast, medication is most often intranasal corticosteroids, which act by reducing mucosal inflammation and require a longer treatment duration. The reversal of developing dentofacial dysmorphology after medical treatment is not well-documented. In addition, comparisons of surgical removal to medical treatment of lymphoid tissue hypertrophy are not available and warrant evaluation.

Accordingly, the aims of this study were to

evaluate, at more than 10 years follow-up, the craniofacial changes in a unique population of children treated for airway obstruction, andcompare long-term orofacial changes in children who had adenoidectomy versus children who were treated medically.

## 2. Materials and Methods

This follow-up investigation included subjects who were previously recruited in a study conducted at [our] university’s medical center to evaluate the effect of mouth breathing on dentofacial structures. The study comprised 280 patients with a mean age of 6.0 years (range: 1.7–12.6 years) who had been referred by the pediatric otolaryngologist to the orthodontic clinics to evaluate adenoid hypertrophy on cephalometric radiographs [8]. The inclusion criterion was the absence of previous orthodontic or surgical craniofacial intervention. Excluded were patients with craniofacial syndromes, cleft lips/palates, and systemic conditions influencing growth.

Upon approval from the institutional review board, the former 280 participants were contacted by phone and introduced to the follow-up study. More than a decade after the original study, only 57 could be recruited, as many had relocated, lacked updated contact information, or declined participation. Inclusion in the present study was considered as acceptance to enroll. The non-participant subjects (‘dropouts’) consisted of 140 (62.78%) males and 83 (37.2%) females, with a mean pre-treatment age of 6.30 + 2.91 years, nearly the same percentage of the participants: 35 (61.4%) males and 22 (38.6%) females with an average age of 5.31 ± 1.99 years.

The new cohort consisted of 35 males and 22 females, with a mean age of 19.09 years (range: 15.1–25.2 years), distributed in a surgical group (n = 34; 21 males, 13 females) and a non-surgical (medically treated) group (n = 23; 14 males, 9 females). In the surgical group, the procedure consisted of an adenoidectomy. In the medical group, patients received intranasal corticosteroids for 3–6 months. The participants were examined after a period of 12.87 ± 1.22 years (range: 10.16–14.76 years). Their mean ages were 5.31 ± 1.99 years at T1 and 18.4 ± 2.03 years at T2 in the surgical group, and 6.77 ± 2.84 years at T1 and 19.4 ± 3.07 years at T2 in the non-surgical group.

Informed consent for participation was obtained from all participants and all the children’s parents. Lateral cephalographs were taken in the same machine used in the initial study (GE, Instrumentarium, Tuusula, Finland) and within the same norms of natural head positioning [18]. The initial T1 and current T2 lateral radiographs of the 57 subjects were imported and digitized into the imaging program (Dolphin Imaging and Management Solutions, Chatsworth, CA, USA). Angular and linear measurements were computed to evaluate the sagittal and vertical positions of the maxilla, mandible and their corresponding dental components, relative to the cranial base and to each other. The measurements included the shortest distance between the adenoid and soft palate (SAD) (Figure 1A).

Considering the wide variation in age at the initial and follow-up assessments, control groups of subjects with Class I malocclusion were established from historical databases, matched for sex and age to the stratified subgroups (surgical, non-surgical) at both T1 and T2. The control images were taken from various sources: the Denver, Iowa, Michigan, and Burlington databases for T1, Bolton and Burlington databases for T2.

All cephalometric measurements were performed by a single calibrated operator, who also repeated the measurements on 20 randomly selected radiographs, 10 from T1 and 10 from T2. The same operator was blinded to treatment group allocation during the T2 assessment to reduce potential bias. T1 and T2 images were de-identified and randomly ordered before tracing, ensuring the operator was blinded to both treatment group and time-point during initial and repeated measurements.

Statistical methods: The intra-class correlation coefficient was computed to determine intra-examiner reliability. The Shapiro–Wilk normality test was run to evaluate the distribution of the data prior to statistical analysis. Paired or independent samples *t*-tests were conducted to compare the means of the different variables between T1 and T2, and between surgical and non-surgical groups and their corresponding matched controls. One-way ANOVA was used to compare the means between surgical and non-surgical groups. SPSS *27.0* statistical software was used to perform all tests; the level of significance was set at *p* < 0.05.

## 3. Results

The intra-class correlation coefficients gauging intra-examiner reliability were greater than 0.9 for all measurements. Representative cephalometric outcomes on patients from the surgical and non-surgical groups with corresponding matched control images are shown in Figure 1.

All measurements at T1 were not statistically significantly different between the surgical and non-surgical groups, except for the following variables that were greater in the non-surgical group: mandibular length (Co-Gn), mandibular body (Go-Pog), and anterior face height (N-Gn). However, these variables were not significantly different at T2 (Table 1).

The gonial angles (Ar-Go-Me), S-N-Me, and N-Gn, as well as SAD were significantly improved in both treatment groups, but changes were greater in the surgical group. The average SNA also decreased significantly in the surgical group and remained nearly unchanged in the non-surgical group (Table 1).

While all T2-T1 changes were statistically significant in both groups, the T2-T1 differences were not statistically significant for the following variables: ANB, OB, OJ, SNA and SNB in the non-surgical group, and OJ and SNB in the surgical group (Table 1).

At T1, SNA and SNB were significantly greater (*p* < 0.001) in the surgical group (86.06° ± 2.94° and 81.71° ± 3.66°) than in corresponding controls (82.15° ± 2.78° and 78.49° ± 2.91°), which did not differ with the non-surgical group (83.0° ± 3.43° and 79.40° ± 3.86°). The inclination of the mandibular plane to the horizontal (MP/HP) and the gonial angle (Ar-Go-Me) were significantly greater (*p* = 0.02 and *p* = 0.01) than the controls (29.49° ± 3.73° and 132.89° ± 4.13°) only in the surgical group (32.2° ± 5.65° and 137.69° ± 8.66°).

Composite renderings of jaw displacements among treatment groups at T2 and the corresponding control groups are illustrated in Figure 2. While SNA values were similar in the surgical (83.43° ± 3.43°) and non-surgical groups (84.57° ± 3.73°), they were greater in the latter than in controls (82.50° ± 1.76°; *p* = 0.02). The same pattern was observed for ANB: 2.61° ± 1.82° and 2.09° ± 0.94° in the surgical and its matched control groups, 3.61° ± 3.54° and 1.87° ± 1.15° in the non-surgical and its matched controls group (*p* = 0.03). PP/MP was significantly different (*p* = 0.02) between the surgical group (23.80° ± 5.72°) and its control (26.84° ± 4.74°), but not between the non-surgical group and its matched control. When compared to control values, the surgical group had a more reduced MP/SN (31.75° ± 5.21° vs. 35.26° ± 3.62°) and gonial angle Ar-Go-Me (125.31° ± 6.43° vs. 129.03° + 5.62°) at *p* < 0.001 and 0.01, respectively.

## 4. Discussion

Two main observations may be drawn from the results of this study:Medical and surgical treatment can be beneficial, as both showed improvement in facial cephalometric features; however, surgery resulted in more improvement in mandibular shape and position. While the mandibular angle (Ar-Go-Me) closed in both treatment groups, the T2-T1 difference was statistically significantly greater in the surgical group (nearly 5°, Table 1). The closure indicates anterior rotation of the mandible and decrease in mandibular plane divergence. The consequent flattening of facial convexity (S-N-Me) was significantly greater in the surgical than the non-surgical group (T2-T1 difference = 4.4°, Table 1), further accentuating the efficacy of surgery (Figure 3).The patients who underwent surgery had more severe facial characteristics and were younger at pre-treatment (5.31 ± 1.99 years vs. 6.77 ± 2.84 years in surgical and non-surgical groups, Table 1). This initial condition may have been the reason for the otolaryngologist to perform and the parents to accept surgery. Treatment selection reflects the protocol followed by the treating otolaryngologist, technically indicating a selection bias, even if unavoidable in reference to the pertinent protocol. Consequently, the differences observed between groups may be attributed not only to the treatment modality but also to the initial severity of the condition, suggesting that severe obstructions are diagnosed early and are eligible for surgical treatment to reach the improvements observed in this study.

Supporting this premise is the finding that medical treatment was apparently implemented because the airway clearance measurement SAD was not severe enough to require surgery: 4.39 ± 2.95 mm at T1, classified as having moderate severity by Macari et al. [19], compared to 2.96 ± 2.45 mm, classified as severe [19] in the surgical group. This measurement was not statistically significantly different between the non-surgical and control groups at T1, while significantly smaller in the surgical group than its corresponding controls (*p* = 0.001). At T2, SAD was more significantly increased in the surgical group than the non-surgical cohort (*p* < 0.001), nearly twice improved in the surgical group, likely because of the adenoidectomy (Table 1).

Moreover, the differential timing of treatment effects between surgical and medical interventions must be considered: adenoidectomy provides immediate relief by physically removing the obstructive lymphoid tissue, thus potentially restoring nasal breathing post-surgically. In contrast, intranasal medication (mainly corticosteroids) reduces mucosal inflammation over a longer treatment duration.

Normalization in maxillary plane rotation (PP/H) in both groups approached the values of the corresponding controls, indicating a trend to normalization of the sagittal and vertical position of the maxilla, more than in the surgical group (Figure 2). Our results do not match those of Woodside et al. [11], who reported no changes in maxillary growth direction after adenoidectomy. This discrepancy may be attributed to the difference in the follow-up periods, 5 years in the latter study, compared to 10–15 years in the present investigation.

Improvement in mandibular length and growth rotation confirmed previously reported mandibular changes following adenoidectomy [9,10]. In a cephalometric study on monozygotic twins of different adenoid dimensions, Dunn et al. [20] reported that the gonial angle decreased with increasing dimensions of the nasopharyngeal airway. Gonial angle reduction was also reported by Mattar et al. [13], who compared the facial effects of post-surgically restored nasal breathing in 33 mouth-breathing young children (aged 3 and 6) with the corresponding features in 22 controls of similar ages during a period of 28 months. In systematic reviews and meta-analysis of surgical interventions, including adenoidectomy, adenotonsillectomy, and tonsillectomy to relieve mouth breathing, reported posttreatment changes included closure of the mandibular plane and decreases in facial height of varying degrees. These results further support our finding of changes in mandibular position and rotation following airway intervention in mouth-breathing children [17,21,22]

Our study further confirms that malocclusions and facial characteristics resulting from environmental factors and associated epigenetic influence can be reversed by timely treatment. The morphological changes that justified treatment of hypertrophied adenoids could be reversed in different proportions. Ghafari et al. [23] suggested that removal before age 6 might lead to more pronounced reversal. While clearance of the airway is reported to positively impact the direction of mandibular growth, [22,23] other findings indicated that children aged 3 to 12.9 years with mild sleep-disordered breathing who underwent adenotonsillectomy experienced improved behavior, enhanced quality of life, and reduced blood pressure at the 12-month follow-up [24]. The findings imply that orthodontists should interact with pediatricians and pediatric otolaryngologists when they determine abnormal or deviant dentofacial development in children with enlarged adenoids and tonsils related to mouth breathing. The presence of dysmorphology might warrant adenoidectomy, at least in patients exhibiting severe features of long-face syndrome that are the least responsive to orthodontic treatment (e.g., a gummy smile associated with anterior open bite and overerupted posterior teeth).

The main strengths of this investigation include the longitudinal follow-up of over 10 to 14.76 years, the longest span to be studied, and the largest (yet modest) sample size in relation to this time interval. In addition, the patients evaluated were past their puberty, therefore, the full growth potential was expressed.

An unavoidable limitation was the inability to recruit more patients to enhance statistical confidence; yet locating nearly a quarter of the original cohort after long periods was remarkable, considering comparable designs in other studies. Nevertheless, potential bias may not be dismissed, particularly that some of the observed changes may also be attributed to normal craniofacial growth over the 13 year interval. Control groups were used to lessen this effect; however, the lack of a contemporary longitudinal matched control group during the study period imposed this methodological limitation, since taking radiographs on such a young population would encounter ethical barriers. This approach is justified under these tenets, notwithstanding the potential for disparities related to geography and ethnicity. A larger sample is warranted for more generalizable conclusions. Also unavoidable was the resort to multiple control databases, in which unknown variables may have been present. Short of an ideal approach with current longitudinal controls, these sources provide validated data commonly used in craniofacial research and as clinical normative guides. Also, to minimize inter-database variability, we limited our selection to untreated Class I subjects matched to each participant.

This study focused on morphological changes through cephalometric measurements. However, other functional parameters related to quality of life, such as comfort in breathing and sleep quality, are clinically relevant and should be included in future studies through pertinent questionnaires and, if possible, objective measures such as polysomnography. Some authors [25,26] reported on the improvement in tongue, lip, and overall orofacial muscle tone, and the maintenance of more potent airways during sleep through functional therapy; however, with a limited base of evidence, further research is needed.

Another limitation is the absence of systematic data on patient compliance with pharmacotherapy. As the effectiveness of intranasal corticosteroids depends heavily on adherence, variability in compliance may have influenced the heterogeneity of outcomes in the medical group. Specifically, the medication may not have been taken regularly or over an optimal period.

## 5. Conclusions

Mouth-breathing children evaluated at an average follow-up visit of 13 years from initial examination exhibited long-term improvements after either adenoidectomy or medical treatment. Adenoidectomy, performed in more severe airway obstructions and at younger ages, was associated with greater reduction in the gonial angle, anterior mandibular rotation, and lower face height. These findings highlight the importance of early recognition and timely management of airway blockage to reduce adverse effects on dentofacial development.

## Figures and Tables

**Figure 1 diagnostics-15-03079-f001:**
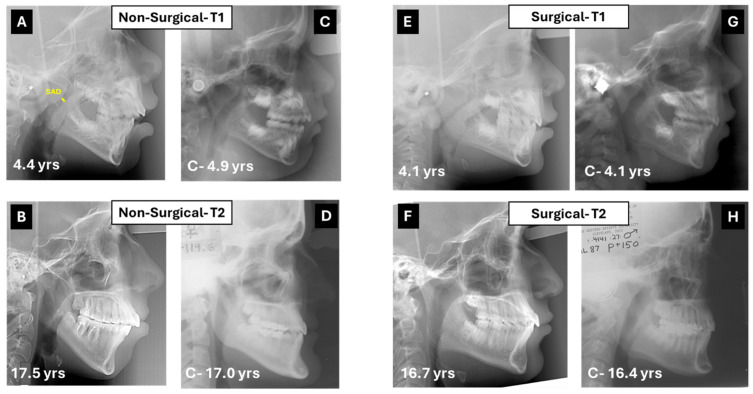
(**A**) Lateral cephalometric radiograph of a 4 year-and-4 months-old girl treated medically for airway obstruction. In yellow, the shortest distance from adenoids to soft palate (SAD). (**B**) Cephalogram of same subject 13 years later. Although the originally posterior caudal inclination of the palatal plane flattened, the hyperdivergent pattern remained with a steep mandibular plane. (**C**,**D**) Control cephalograms from the Michigan Growth Study (**C**) and the Bolton–Brush Growth Study (**D**) matched for sex and age with corresponding cephalograms at T1 and T2. (**E**) Lateral cephalograph of a 4.1-year-old boy treated with adenoidectomy. Note the hyperdivergent pattern with posterior–inferior tip of the palatal plane. (**F**) The cephalogram taken 14 years later reveals a shift to normal divergence of the jaws. (**G**,**H**) Control cephalograms from the Burlington Growth Study (**G**) and the Bolton–Brush Growth Study (**H**) matched for sex and age with corresponding cephalograms at T1 and T2.

**Figure 2 diagnostics-15-03079-f002:**
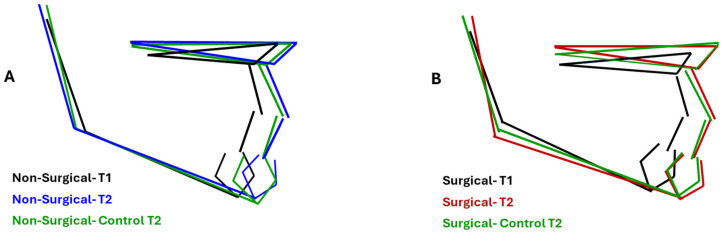
Comparison of composite rendering of maxillary and mandibular displacement in surgical, medication, and corresponding control. (**A**) Medication and control groups were similar at T2. (**B**) The gonial angle closed at T2 in the surgical group compared to T1 and was more acute than the control group. Mandibular length was also greater than the control group at T2.

**Figure 3 diagnostics-15-03079-f003:**
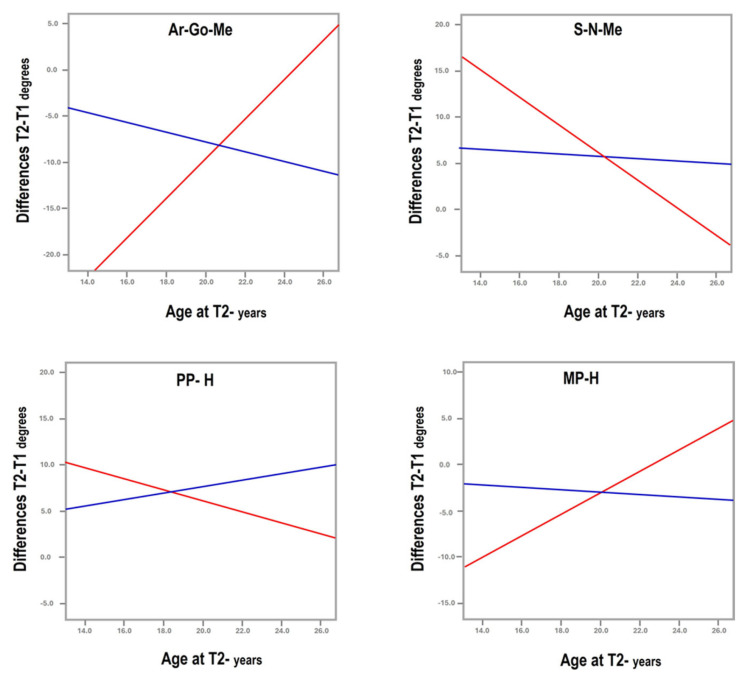
Graphs illustrating the deviating curves between surgical (red) and non-surgical (blue) treatment of airway obstruction for the gonial angle (Ar-Go-Me), angle of convexity (S-N-Me), and maxillary (PP/H) and mandibular (MP/H) inclinations to the horizontal.

**Table 1 diagnostics-15-03079-t001:** Comparison of cephalometric measurements of each group at T1 and T2 and their mean differences (T2-T1).

	Non-Surgical Group N = 23					Surgical Group N = 34					Non-Surgical		Surgical		
	T1		T2		Sig.	T1		T2		Sig.	T2-T1 Diff.		T2-T1 Diff.		Sig.
	Mean	SD	Mean	SD		Mean	SD	Mean	SD		Mean	SD	Mean	SD	
Age (years)	6.77	2.84	19.4	3.07		5.31	1.99	18.4	2.03		12.63	2.86	13.09	2.11	
** *Skeletal Measurements* **															
ANB°	4.40	2.88	3.61	3.54	0.20	4.47	2.39	2.61	1.82	0.00	−0.79	2.84	−1.86	2.00	0.10
PP-MP°	30.00	4.81	25.66	7.18	0.00	29.37	4.39	23.80	5.72	0.00	−4.34	5.39	−5.57	5.48	0.41
LFH/TFH	56.84	2.27	55.52	2.08	0.05	57.43	2.05	55.96	2.66	0.00	−1.32	3.06	−1.47	2.51	0.97
SNA°	84.67	3.45	84.57	3.73	0.92	86.06	2.94	83.43	3.43	0.00	−0.10	4.81	−2.63	3.31	0.02
ANS-PNS mm	43.41	3.52	52.83	3.93	0.00	42.01	3.04	52.07	3.84	0.00	9.42	4.95	10.06	4.08	0.59
PP-H°	−5.20	4.40	2.47	2.60	0.00	−5.23	5.76	2.23	2.53	0.00	7.67	4.75	7.46	5.40	0.89
SNB°	80.26	3.13	80.97	3.49	0.46	81.71	3.66	80.85	3.57	0.17	0.71	4.48	−0.86	3.56	0.15
MP-SN°	34.16	5.38	31.14	5.07	0.01	34.29	2.59	31.75	5.21	0.00	−3.02	4.70	−2.54	4.64	0.70
S-N-Me°	57.37	4.84	63.80	4.12	0.00	55.02	5.15	64.85	3.48	0.00	6.43	4.73	9.83	6.32	0.03
Co-Gn mm	91.46	7.48	115.58	10.63	0.00	87.12	8.01	115.16	8.36	0.00	24.12	10.15	28.04	9.55	0.14
Ar-Go-Me°	133.90	7.83	126.66	6.42	0.00	137.69	8.66	125.31	6.43	0.00	−7.24	4.97	−12.38	8.25	0.01
N-Gn mm	98.82	8.55	119.49	8.76	0.00	93.30	8.96	120.06	7.29	0.00	20.67	8.21	26.76	10.07	0.02
** *Dentoalveolar Measurements* **															
U1/NA°	15.06	7.58	22.37	8.77	0.01	13.90	7.52	25.52	5.56	0.00	7.31	12.09	11.62	9.61	0.14
U1/PP°	102.87	7.54	111.98	6.82	0.00	102.78	7.45	115.59	5.64	0.00	9.11	11.48	12.80	9.50	0.19
L1/NB°	23.86	6.20	26.44	7.15	0.03	23.39	3.18	27.09	5.83	0.00	2.58	0.95	3.7	2.65	0.17
L1/MP°	89.90	6.73	93.63	8.81	0.02	89.01	4.52	94.92	7.62	0.00	3.73	6.79	5.91	7.57	0.27
OB mm	0.49	2.45	0.97	1.79	0.26	0.39	1.77	1.24	1.61	0.03	0.48	2.00	0.85	2.18	0.52
OJ mm	2.93	2.26	3.33	1.64	0.37	2.88	2.18	3.34	1.74	0.34	0.40	2.12	0.46	2.73	0.94
** *Airway* **															
SAD mm	4.39	2.95	11.84	2.81	0.00	2.96	2.49	16.25	2.94	0.00	7.45	3.31	13.29	3.22	0.00

## Data Availability

The raw data supporting the conclusions of this article will be made available by the corresponding authors on request.

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
