# Peer review of "Long-Term (>10 Years) Effects of Medical and Surgical Airway Obstruction Treatment on Dentofacial Morphology"

_diagnostics, 2025, doi:10.3390/diagnostics15233079_

Round 1
Reviewer 1 Report (New Reviewer)
Comments and Suggestions for Authors
Commendable Longitudinal Design
The extended follow-up period of over 10 years is a major strength of this study. It provides valuable insights into the sustained impact of airway obstruction treatments on dentofacial development, which is rarely captured in existing literature
Clinical Relevance and Implications
The study highlights the importance of early diagnosis and intervention for airway obstruction. Consider expanding on how these findings could inform interdisciplinary collaboration between orthodontists and ENT specialists in clinical practice.
Major issue
Suggestions for Future Research
While the morphological outcomes are well-documented, future studies could benefit from incorporating functional assessments such as sleep quality, breathing comfort, or patient-reported outcomes. This would provide a more holistic view of treatment impact.
Limited Sample Size and Attrition
Only 57 out of 280 original participants were included in the follow-up, raising concerns about representativeness and statistical power. The authors should discuss how attrition may have influenced the results and whether the final sample reflects the broader population.
Personal Relevance and Clinical Perspective
I appreciate the study’s focus on long-term dentofacial outcomes. It would be valuable for the authors to discuss how surgical interventions like adenotonsillectomy compare with orthodontic or functional therapies in terms of airway stability and craniofacial development. Including recent literature that explores multidisciplinary approaches—such as the 2025 review in Oral (DOI: 10.3390/oral5030055)—could help contextualize these findings within current clinical practice.
Author Response
Please see the attachment.

Reviewer 2 Report (New Reviewer)
Comments and Suggestions for Authors
This manuscript addresses a significant and clinically relevant gap in the literature: the very long-term (>10 years) dentofacial morphological changes following treatment for paediatric airway obstruction, comparing surgical (adenoidectomy) and medical (intranasal corticosteroids) interventions. The methodology is generally sound, and the results are compelling. However, several areas require clarification and refinement to enhance the manuscript's impact and scientific rigor.
Below are my comments:
Materials and Methods.
Page 2, line 82. Upon approval from the institutional review board, the former 280 participants were contacted by phone and introduced to the follow-up study. More than a decade after the original study, only 57 could be recruited, as many had relocated, lacked updated contact information, or declined participation. Only 57 of 280 original participants were retained. This introduces significant selection bias. The authors should report characteristics of dropouts (e.g., initial age/sex distribution) and discuss representativeness.
Page 3, line 119. The control images were taken from various sources: the Denver, Iowa, Michigan, and Burlington databases for T1, Bolton and Burlington databases for T2. Justify why this approach is valid and how inter-database variability was minimized?
Page 3, line 121. All cephalometric measurements were performed by a single calibrated operator who also repeated the measurements on 20 randomly selected radiographs, 10 from T1 and 10 from T2. The same operator was blinded to treatment group allocation during the T2 assessment to reduce potential bias. While the operator was blinded to treatment group, it is unclear whether image order (T1 vs. T2) was blinded, this can affect subjective tracing.
Results.
Page 4, Table 1. The data presentation is overwhelming. The table is dense. The authors should highlight clinically significant variables (gonial angle, facial height, S-N-Me, SAD) instead of listing all.
Discussion.
Page 8, line 218. Improvement in mandibular length and growth rotation confirmed previously reported mandibular changes following adenoidectomy. The discussion references older works (Linder-Aronson, 1986 and Kerr WJ, 1989). Consider integrating more recent meta-analyses or imaging-based growth studies.
Page 8, line 242. An unavoidable limitation was the inability to recruit more patients to enhance statistical confidence; yet locating nearly a quarter of the original cohort after long periods was remarkable considering comparable designs in other studies. The limitation of low recruitment is stated too mildly. A ~80% loss to follow-up is a major issue that potentially biases the results. This needs to be discussed more robustly, acknowledging the potential for non-response bias.
Page 8, line 246. Control groups were used to lessen this effect; however the lack of a contemporary longitudinal matched control group during the study period imposed this methodological limitation, since taking radiographs on such a young population would encounter ethical barriers. The limitation of the control group is well-stated, but it should also explicitly mention the potential ethnic/geographic mismatch, which is a significant confounder for cephalometric.
Page 8, line 254. Another limitation is the absence of systematic data on patient compliance with pharmacotherapy. As the effectiveness of intranasal corticosteroids depends heavily on adherence, variability in compliance may have influenced the heterogeneity of outcomes in the medical group. The limitation of non-compliance is critical. It's possible the "medical treatment" had a lesser effect simply because compliance was low, or because the 3-6 month duration was insufficient for a condition that may have persisted. This uncertainty must be emphasized.
Data Availability Statement.
Page 9, line 271. Data Availability Statement: We encourage all authors of articles published in MDPI journals to share their research data. In this section, please provide details regarding where data supporting reported results can be found, including links to publicly archived datasets analyzed or generated during the study. Where no new data were created, or where data is unavailable due to privacy or ethical restrictions, a statement is still required. Suggested Data Availability Statements are available in section “MDPI Research Data Policies” at https://www.mdpi.com/ethics. Please update this data availability statement.
The manuscript is scientifically relevant and contributes valuable long-term data. However, it requires enhanced clarity in methodology and statistics, a more balanced interpretation of results, and improved focus and conciseness in the abstract and discussion. I look forward to seeing a revised version of this manuscript.
Round 2
Reviewer 1 Report (New Reviewer)
Comments and Suggestions for Authors
The authors have successfully addressed all of my comments. I have no further comments.
Reviewer 2 Report (New Reviewer)
Comments and Suggestions for Authors
The authors have adequately addressed all comments
This manuscript is a resubmission of an earlier submission. The following is a list of the peer review reports and author responses from that submission.
Round 1
Reviewer 1 Report
Comments and Suggestions for Authors
Dear Authors,
This article addresses the important issue of examining the long-term (>10 years) effects of medical and surgical treatments on craniofacial morphology in children with airway obstruction due to mouth breathing. However, there are several significant shortcomings and uncertainties that undermine the scientific validity of the article and the reliability of its findings.
1. Deficiencies in the Methods and Data Collection Section
The methods section of the article lacks some critical information necessary to ensure the reproducibility of the study and the validity of the results.
Patient Selection Criteria: Only 57 of the initial study population of 280 patients were contacted. This is approximately 20% of the original cohort. Insufficient information was provided as to why the remaining 80% were not contacted or included in the study. This could lead to selection bias. It is possible that the patients contacted had different characteristics than those not contacted, making it difficult to generalize the results to the entire patient population.
Treatment Protocols: Details such as the type of medication used (e.g., intranasal corticosteroids), dose, and duration of treatment in the medical treatment group are not specified. Similarly, details of the adenodectomy performed in the surgical treatment group (e.g., whether it was accompanied by tonsillectomy) are unclear. This lack of information reduces the robustness of comparisons between treatment groups.
Baseline Differences Between Groups: The article states that the surgical group had more severe anomalies at baseline (larger gonial angle, smaller SAD). This suggests that the treatment choice was already made in favor of surgery for more severe cases and medical treatment for less severe cases. This is a significant confounding factor, making it difficult to assert that the two groups were not initially homogeneous and that the difference between them was due solely to the type of treatment.
Use of Control Groups: While the study used control groups, the fact that these groups were compiled from older growth studies (Denver, Iowa, Michigan, Burlington, and Bolton-Brush) is a significant methodological limitation. These control groups were not followed for the same period as the patients in the article and are not contemporary controls. This weakens the comparison due to factors such as the fact that growth patterns in different populations may differ from those in the current population and that radiological techniques may change over time.
2. Uncertainties in the Results and Discussion Section
The results section of the article contains logical gaps between the presented data and the conclusions drawn, and the discussion section fails to adequately fill these gaps.
Statistical Analysis Comment: In Table 1, statistically significant improvement (p<0.05) was observed in many measurements between T1 and T2 in both the surgical and non-surgical groups. However, the article emphasizes that the surgical group showed greater improvement. In the column showing the comparison of the difference in change between the two groups (T2-T1 diff), only one measurement (S-N-Me) showed a statistically significant difference (p=0.03). Although a significant difference in favor of the surgical group is claimed for the gonial angle (Ar-Go-Me) (p=0.01), this result is less emphasized compared to the other variables mentioned in the article. This suggests an inconsistency in the interpretation of the results.
Effect of Age at Treatment: The article states that the surgical group was treated at a younger age (mean 5.31 years at T1). The discussion section addresses the hypothesis that earlier intervention may yield better outcomes. However, no statistical analysis (e.g., regression analysis) was conducted to demonstrate the relationship between age at treatment and morphological improvement. Therefore, it is unclear whether the observed differences are due to the surgical intervention itself or to the intervention performed at a younger age.
Effect of Growth: While the article presents the fact that the patients were past puberty as a strength, it should not be overlooked that the 13-year period between T1 and T2 may also cause morphological changes as part of normal growth and development. While the use of control groups attempts to control for this effect, the robustness of this comparison is questionable due to the methodological limitations mentioned above.
3. Overall Assessment of the Article
While this article is an important study examining the long-term effects of airway obstruction treatment on dentofacial morphology, it is difficult to draw strong conclusions due to key methodological shortcomings. The article does not adequately consider confounding factors such as the surgical group having more severe conditions at baseline and being treated at an earlier age.
Reviewer 2 Report
Comments and Suggestions for Authors
Thank you for the article titled "Long-Term (>10 Years) Effects of Medical and Surgical Airway Obstruction Treatment on Dentofacial Morphology." This article addresses a significant clinical issue with considerable practical and scientific implications. However, I believe it would benefit from additional information.
In the Introduction section, the authors could briefly explain the differences between the treatment methods. For instance, adenoidectomy provides immediate relief by removing the physical obstruction, while pharmacotherapy offers anti-inflammatory effects, involves a longer treatment duration, and may result in less stable outcomes.
In the Materials and Methods section, the authors stated that 280 children were invited to participate in the study, but only 57 ultimately enrolled. However, there is a lack of information regarding the inclusion and exclusion criteria for the study participants. Additionally, it remains unclear whether the children who did not return for follow-up differed significantly from those who were included in the study, which raises concerns about potential selection bias.
While the characteristics of the groups are presented, further clarification is needed. Specifically, it would be helpful to provide details about the severity of obstruction, the type of pharmacological treatment administered, and the duration of that treatment to enhance understanding.
It should also be clarified whether all radiographs were analyzed by a single operator or by multiple operators, as inter-examiner agreement is necessary. Furthermore, it is essential to specify whether the measurements were taken without knowledge of the participants' group assignments to prevent investigator bias.
There is also insufficient information regarding the informed consent obtained from the patients or their parents or guardians.
In the Results section, the table is extensive and contains some repeated data in the text. In my opinion, it's important to remove the repetition in the text.
In my opinion the Discussion requires some additions. The authors note that the surgical patients were younger and had more severe obstruction; however, the potential for selection bias should be emphasized more strongly, as this could partially explain the more pronounced effects of surgical treatment.
The discussion primarily focuses on the cephalometric results. However, there is a lack of commentary regarding the study's omission of functional aspects, such as quality of breathing, sleep, and overall quality of life, which are clinically relevant. This absence of functional assessment is an important limitation that should be acknowledged.
In the pharmacologically treated group, there is insufficient data on patient compliance. The effectiveness of pharmacotherapy is highly dependent on the regular use of medication, and this should be noted.
The Conclusion section should be more concise, focusing on the clinical significance of the results obtained.